# Senses by Seasons: Tourists' Perceptions Depending on Seasonality in Popular Nature Destinations in Iceland

**Anna Dóra Sæþórsdóttir [1],\*, C. Michael Hall [2,3,4]**  **and Þorkell Stefánsson [1]**

[1] School of Engineering and Natural Sciences, University of Iceland, Askja, Sturlugata 7, 101 Reykjavík, Iceland; ths33@hi.is

[2] Department of Management, Marketing and Entrepreneurship, University of Canterbury, Private Bag 4800, Christchurch 8140, New Zealand; michael.hall@canterbury.ac.nz

[3] Geography Research Unit, University of Oulu, FI-90014 Oulu, Finland

[4] School of Business & Economics, Linnaeus University, 391 82 Kalmar, Sweden

\* Correspondence: annadora@hi.is

**Abstract:** Seasonality in visitor arrivals is one of the greatest challenges faced by tourist destinations. Seasonality is a major issue for sustainable tourism as it affects the optimal use of investment and infrastructure, puts pressure on resources and can create negative experience of crowding at destinations. Peripheral areas commonly experience more pronounced fluctuations in visitor arrivals. Iceland is one of those destinations. Although the number of tourists visiting the country has multiplied in recent years, seasonality is still a major challenge, especially in the more rural peripheral areas of the country. Iceland's high season for tourism occurs during its brief summer (June to August), but in recent years more people visit the country on shorter winter trips, creating new management challenges. This research is based on an on-site questionnaire survey conducted in seven popular nature destinations in Iceland which compares the experience of summer and winter visitors. The results show that winter visitors are more satisfied with the natural environment while their satisfaction with facilities and service is in many cases lower. The areas are generally perceived as being more beautiful and quieter in winter than in summer. However, most destinations are considered less accessible and less safe in the winter. Tourists are much less likely to experience physical crowding during winter, although winter visitors are more sensitive to crowds, most likely because of expectations of fewer tourists. Finally, this research shows that tourists are less likely to encounter negative effects of tourism on the environment in the winter, (e.g., erosion or damage to rocks and vegetation), than in summer. The results highlight the importance of understanding visitor perceptions in a seasonal and temporal context.

**Keywords:** tourism seasonality; tourists' perception; crowding; sustainable management

## 1. Introduction

Seasonality, with respect to fluctuations in visitor arrivals, is one of the major characteristics of tourism and is widely regarded as one of its major problems [1–3]. Seasonality in tourism usually involves a higher number of visitors at a destination for a specific period of a year [4,5], with fewer visitors at other times. Seasonal demand occurs when the pattern of demand over time is characterized by regular fluctuations that are relatively consistent with the pattern of supply, although excess capacity to supply the product will exist at certain times of the year. Seasonal demand is different from irregular demand [6] which is "A state in which the current timing pattern of demand is marked by seasonal or volatile fluctuations that depart from the timing pattern of supply" [7]. Tourism seasonality can

therefore be defined as the regularly occurring fluctuations in the supply and demand of tourism, which may be identified in terms of such factors as regular changes in tourist numbers and/or their consumption, expenditure, transport flows, tourist sector employment, accommodation availability, and resource use. This imbalance has historically affected the optimal use of tourism infrastructure and resources and therefore its potential economic and employment contributions [8–12]. Consequently, the reduction of seasonality, usually as expressed by the gap between high and low levels of tourist demand for a particular product or destination, has long been one of the major issues in tourism management [1] and has often been dealt with as a significant policy issue [13,14].

Natural and institutional elements strongly influence seasonality. Hylleberg [15] identified three categories of seasonality (i.e., weather, calendar effects, and timing decisions). Natural elements, such as season changes in climate and daylight have a great influence on tourism. Institutional elements, such as official public holidays and religious and/or cultural holidays, are also very influential, along with legal requirements for holidays under employment law, and affect the holiday travel decisions of leisure tourists [1,3,16,17].

Houghton et al. [18] identified three major peak patterns for destinations, those being single peak, double peak, and non-peak destinations. Double peak destinations experience two periods of higher demand than other times of the year. This situation is often the case for alpine resorts that experience peak demand in summer and winter [18,19]. Non-peak destinations are often less influenced by natural and climatic elements. These can include cities, which usually demonstrate a more diversified demand and have transport systems and large indoor attractions that are not affected by external climate, and therefore operate without a significant seasonal peak [20,21]. Seasonality is also influenced by a destination's location, and relative accessibility within a country or region [1,22–24]. There is a suggestion that seasonality may be relatively more prominent at destinations in peripheral areas as they usually experience more pronounced fluctuations [25–27], and consequently shorter periods in which to generate returns from tourism. Climatic elements, such as ice, snow, or rainfall can also affect the accessibility of remote locations during off-season periods and can be a major barrier to shifting visitation patterns [1,24,28,29].

Seasonality in tourism has substantial implications for the financial sustainability of businesses and the broader environmental, economic, and social sustainability of destinations. A short high season means either under-utilization of investments in the low season, and thereby limited return on capital, or over-capacity during the high season [30,31]. Seasonality also affects employment stability, mainly due to difficulties in hiring and retaining staff [32,33], and is therefore a major contributor to the short-term and casual nature of much tourism employment as well as firm productivity [11]. Such a situation also often means that the overall economic benefits of tourism to destinations may be substantially limited with seasonal employment taken up by people from outside the region [34], the arrival of whom may also put pressure on housing and rental markets [35]. Many countries, such as Australia and New Zealand [36], even offer short-term working-holiday and employment visas as a means of assisting the supply of short-term staff to seasonal business operations. The growth of tourist and second home visits during the peak season can also have negative environmental impacts particularly if infrastructure provision, for waste and sewage for example, is based on that of permanent populations [2,37,38]. Additionally, the negative experience of crowding has long been recognized as a consequence of seasonality for destination residents, as well as visitor satisfaction and the quality of the visitor experience [31,39,40]. For local people, problems may include congestion and slower traffic, limited parking spaces, longer wait times, and higher prices for services [41,42]. On the other hand, many facilities and services may close during the off-season [43], which can lower the reputation and image of the destination [44].

However, despite a range of research on tourism and seasonality, the majority of studies have been undertaken in urban or relatively developed destinations. Research concerning attitudes of tourists and their satisfaction as a result of seasonality in natural landscapes appears very limited [45–47], along with assessments of seasonal variations in the types of visitors [47–49]. Such understandings



of the interrelationships between seasonality, landscape perception, and tourist behavior are also becoming increasingly important given the impacts of climate change on seasonal landscapes and their access and attractiveness to tourists [26,50–53].

This paper examines seasonal variation in seven nature destinations in Iceland by comparing attitudes and characteristics of summer and winter visitors. Following an overview of tourism seasonality in Iceland, the paper reports the results of a survey that analyzes visitors' perception of crowding and the environment (i.e., regarding its naturalness, beauty, cleanliness, quietness, accessibility, and safety). Additionally, it presents visitors' satisfaction with the stay, nature, infrastructure, and service.

## 2. Tourism Seasonality in Iceland

Iceland is an example of a destination characterized historically by substantial seasonality. Since the first formulations of a national tourism policy in 1975 [54], one of the main aims of Icelandic tourism policy has been to even seasonal fluctuations in international visitor arrivals [55,56]. However, despite ambitious aims to limit seasonality there had not been much success until lately.

Iceland has experienced an enormous increase in the overall number of international tourist arrivals to the country since 2010. In 2010 the number of international visitors to Iceland was about 460,000, while in 2018 it had gone up to approximately 2.3 million [57]. That is almost a fivefold increase or an annual average increase of about 22% between 2010 and 2018. At the same time seasonality has decreased, with the proportion of tourists arriving in the three summer months (June, July, and August) dropping from 50% in 2010 to 35% in 2018. The greatest reduction in seasonality has occurred in the past few years; between 2014 and 2017 seasonality evened out considerably, with the greatest increases in monthly arrivals being outside of the summer period (Figure 1). For example, between 2016 and 2017 there was a 75% increase in visitors in January and 62% in April while the average growth in the summer months was 17%. In 2018 overall growth slowed substantially to 5%, with the largest increase occurring in May and September [57].

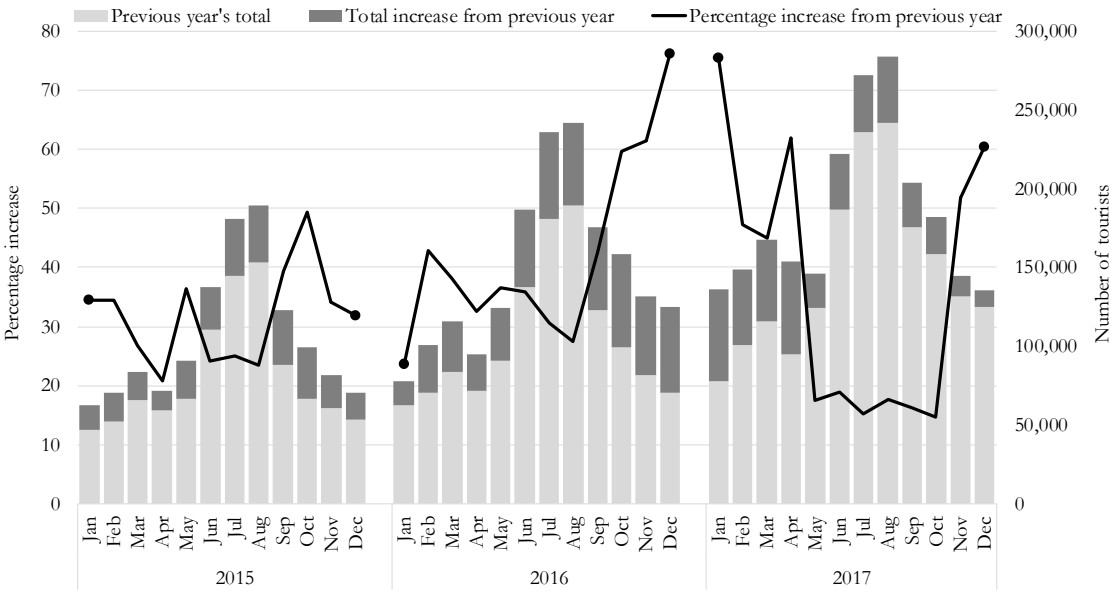

**Figure 1.** Seasonal changes in international visitor arrivals 2015–2017 [57].

Several reasons may be offered for the reduced seasonality in Icelandic tourism, including the increased focus on the marketing of winter tourism by the government and industry through initiatives such as "Iceland all year-round" [58,59]. Another driver is foreign airlines' decision to include Iceland as a year-round destination resulting in lower fares during the off-season. According to visitor surveys from 2015–2016, about 36% of foreign visitors named low airfares as a major driver in the decision to

visit Iceland during the winter, while in the summer the figure is between 17% and 19% [60]. Meanwhile, the strength of nature as an attraction for tourists during winter has also been increasing [60].

Accordingly, fluctuations in overnight stays have been diminishing along with the increase in off-season tourism, but this is occurring to a far greater extent in the capital area than in other regions (Figure 2). In 2010 half of all overnight stays in the capital area were in May–August, while in other regions, 87% of all overnight stays were in that period. In 2017 this proportion changed to 36% in the capital and 61% in other regions [61].

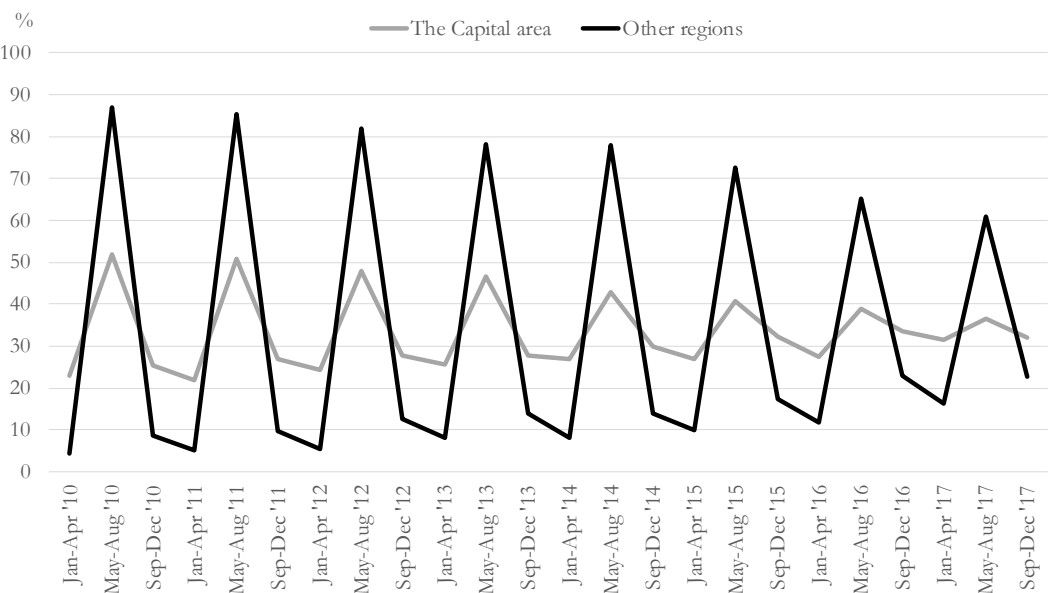

**Figure 2.** Proportion of overnight stays in the capital area and other regions in each three-month period from 2010–2017 [61].

Another example of the general reduction of seasonality is evident in the change of the occupancy rate of hotels from 2010 to 2017. As an example, the country's overall occupancy rate in February increased from 38% in 2010 to 78% in 2017. However, the increase in occupancy rates has mostly been limited to the southern part of the country. Occupancy rates in the capital area have risen from 49% to 96% in February and from 75% to 93% in July (Figure 3). An even more dramatic increase has occurred in the Southwest, where the international airport is located. There, the occupancy rate went from 32% to over 90% in February and from 67% to 90% in July. The South has also had a substantial change in increased occupancy rates, especially during winter as it has gone from 26% to 68% in February. The East and the North, the areas furthest from the capital area, still had the lowest occupancy rates in February, about 17% and 30%, respectively, although this figure has risen since 2010. In July of 2017 all regions had an occupancy rate of about 80% or higher [61].

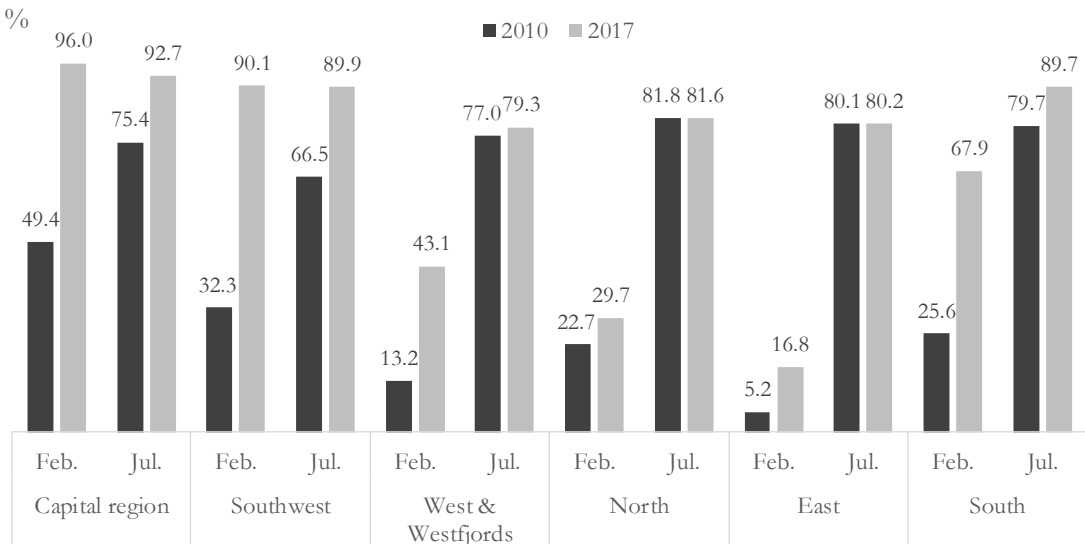

**Figure 3.** Occupancy rate of hotels in February and July between 2010 and 2017 [61].

Therefore, even though these data show a high level of success in limiting seasonality, it has so far mostly benefited the southern part of the country. One reason for this could be that the length of stay is shorter during the off-season. During the winter 79% of foreign tourists stay in Iceland for seven days or less, but during summer only 42% stay for a week or less while the majority stays longer [60,62]. Since almost 99% of all visitors to Iceland arrive through the main international airport at Keflavík [57], which is located on the southwest peninsula Reykjanes, about a 30 minute drive from the capital area (Reykjavík), the majority (97%) also visit and stay overnight in the capital region. The time constraints of shorter stays may limit people's opportunities to visit and stay overnight in places further away from the capital, so a pattern of going on day tours to nature destinations in the south and southwest corner of the country is prevalent.

## 3. Methods

### 3.1. The Research Areas

An on-site questionnaire survey was conducted to assess tourist experiences at seven popular nature tourist destinations in Iceland in the summer of 2014 and winter of 2015. The areas selected for the study were Geysir, Þingvellir, Jökulsárlón, Sólheimajökull, Seltún, Hraunfossar, and Djúpalónssandur (Figure 4). These were selected as they are among the most visited nature destinations of the country. Two of these, the Geysir hot spring area and the Þingvellir national park, which is also a UNESCO World Heritage Site, are core offerings in the country's most popular excursion called "The Golden Circle". All seven research areas are located in the southwest of the country and can be reached on a daytrip from the capital area. Jökulsárlón is located furthest away from Reykjavík (379 km) and is about a 12–15 h round daytrip from Reykjavík.

According to a survey among tourists leaving the country at Iceland's main international airport in Keflavík, 60% of international visitors came to Geysir in the summer of 2014 and 41% in the preceding winter, while 50% visited Þingvellir in the summer and 33% in the winter. About 42% visited the glacial lagoon Jökulsárlón in the summer and 21% in the winter [63,64]. The other sites, Djúpalónssandur, Hraunfossar, Seltún, and Sólheimajökull are not quite as popular as the others, although they are still highly visited.

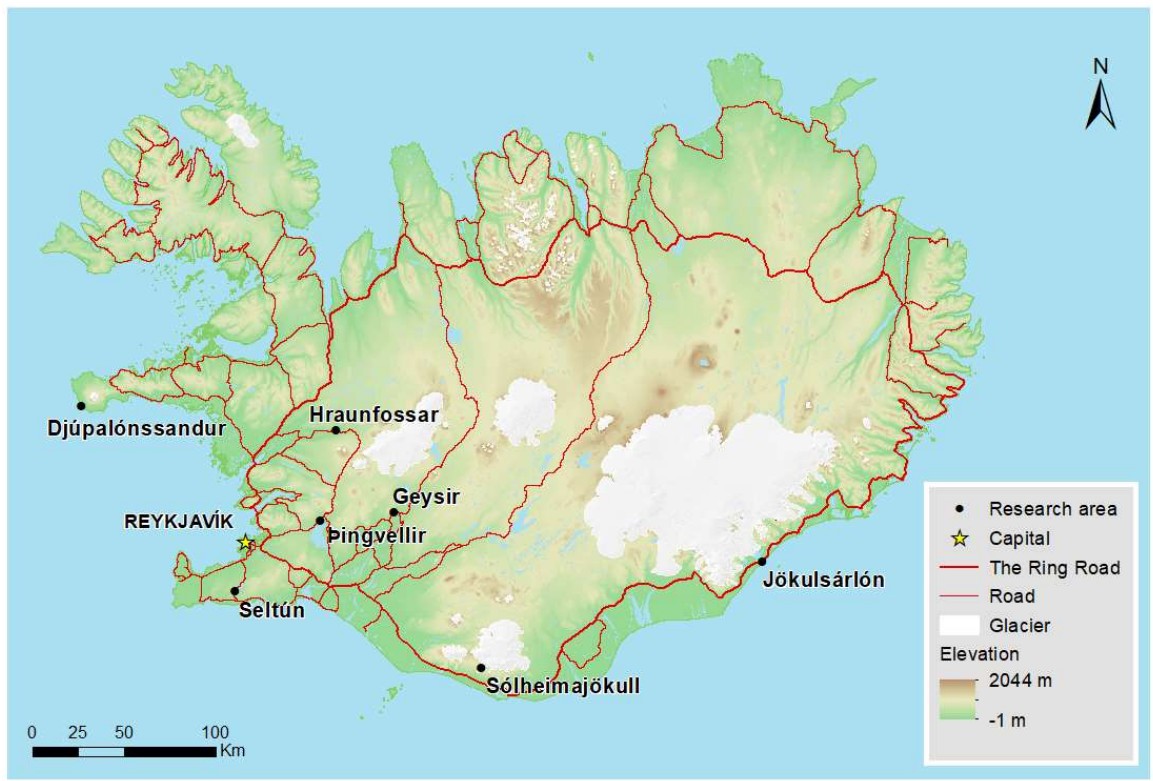

**Figure 4.** The seven research areas and Reykjavík, the capital city of Iceland.

All research areas except Geysir were included in a 2015 study by Þórhallsdóttir and Ólafsson (2017) which counted the number of visitors (Figure 5). Out of the six remaining destinations, Þingvellir was the most visited with 697,000 annual visitors, followed by Jökulsárlón with 450,000. Sólheimajökull had 201,000 visitors, Seltún received 136,000 guests, and 94,000 people visited Djúpalónssandur in 2015.

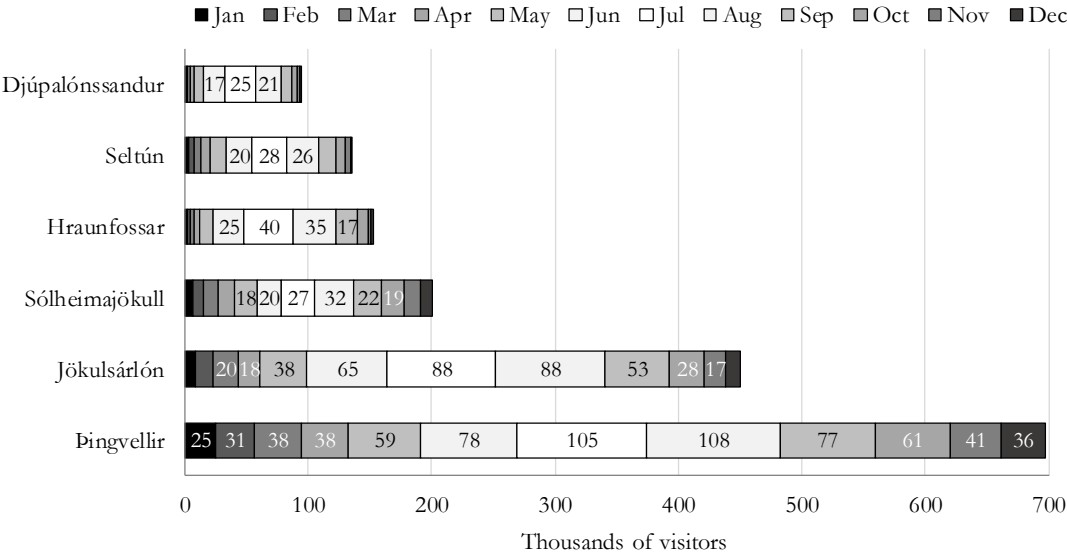

**Figure 5.** Number of visitors by months in 2015 at six of the research areas [65].

The most visited areas, Geysir and Þingvellir, have infrastructure available year-round, while some of the other places have more limited services which are not available the whole year. The infrastructure at Geysir and Þingvellir has been developed in recent years but in somewhat different ways. Due to distinct geographical features, the infrastructure and facilities at Geysir are situated close to the

geothermal area itself. In fact, Geysir has been put on a so called "red list" of places where the environment is in danger, in this case due to tourism's impact in the area [66]. At Þingvellir, on the other hand, the infrastructure has been placed at the entrance of the national park, making it less visible once visitors have entered the park. Jökulsárlón has more limited and basic infrastructure, despite the high number of visitors.

Iceland is an island in the North Atlantic Ocean, located between 63° and 66° northern latitudes and lies in the path of the warm Gulf Stream and consequently the temperature is moderate given its latitude. The average temperature in winter is around 0 °C (32 °F) in the southern lowlands. The summers are cool, the average temperature in the south being around 10–13 °C (50–55 °F). The weather in Iceland can be particularly variable. During the seven weeks of data gathering in the summer, rain was experienced at all locations. In fact, July 2014 was the wettest for a decade in many areas of the country but wind speed was close to average [67]. The temperature in February 2015 was slightly lower than the average in the south and southwest. The month was also wetter and less sunny than on average [68]. All the destinations had a thin white snow cover during the week of data collection in the winter. In addition, in the beginning of February there was about 8 h of daylight available compared to 20 h in July.

### 3.2. Data

The target population of this study were all visitors at popular nature destinations in Iceland. Devising an effective sampling frame can be problematic in on-site tourism research in natural areas. Accessing the entire population is extremely difficult as the study area is often very large or parts of it are inaccessible. It was obviously unrealistic to gather information from the entire visitor population in large natural areas, and therefore a sample had to be taken. Seven popular nature destinations were selected that represent a wide range of environments and use levels. Data was collected by one to five interviewers, depending on the destination's previously reported popularity (see [65]). The interviewers stayed at each destination for a week. A week-long stay was chosen at each location as some visits follow a day-of-the-week pattern. For example, the number of Icelandic visitors is higher at the weekend, while some organized group tours have fixed days of arrival each week. A self-completion survey was given out during the day approximately between 9:00 a.m. and 19:00 p.m. The utilization of self-completion questionnaires required less staff to gather the data and made it possible to get answers from tourists on buses. Primarily, a simple random sampling was used where the interviewers attempted to approach as many visitors as possible. However, organized tourist groups often do not have time to answer the questionnaire on site as their visit is part of a set tour timetable. They can therefore easily be under represented by a simple random sample approach. Therefore, in order to avoid bias, a stratified sampling was used for tour groups with travel guides or bus drivers being asked to distribute the questionnaire among the visitors on the bus. Those who agreed were given postage-paid envelopes and were asked to return them via mail after completion.

A total of 18,957 questionnaires were collected, 12,199 in the summer and 6758 in the winter. The largest sample was collected in the most visited area in the summer (17,600 surveyed at Þingvellir) and the smallest in a less visited area (356 at Djúpalónssandur) in the winter (Table 1). Another study was being undertaken in the same locations during the same period evaluating the number of visitors, also financed by the Icelandic Tourist Board [65]. By using the data from that study an approximate response rate to the survey presented here can be calculated. The response rate in the summer was between 15% and 37% and in the winter between 11% and 37% (Table 1).

**Table 1.** Dates of data collection, sample size, and response rate.

| | Summer 2014 | | | | Winter 2015 | | | | |
|---|---|---|---|---|---|---|---|---|---|
| | Data Collection (dd.mm) | Sample Size (n) | No. of Visitors (N) * | Resp. Rate | Data Collection (dd.mm) | Sample Size (n) | No. of Visitors (N) * | Resp. Rate | Total Sample Size |
| **Djúpalónss.** | 23.06–29.06 | 793 | 2922 | 27 | 02.03–08.03 | 132 | 356 | 37 | 925 |
| **Geysir** | 05.08–11.08 | 2868 | n/a | n/a | 02.03–08.03 | 2421 | n/a | n/a | 5289 |
| **Hraunfossar** | 09.07–14.07 | 1419 | 5712 | 25 | 17.03–23.03 | 360 | 1046 | 34 | 1779 |
| **Jökulsárlón** | 18.07–22.07 | 2056 | 14,003 | 15 | 23.02–01.03 | 474 | 4441 | 11 | 2530 |
| **Seltún** | 12.08–18.08 | 934 | 4795 | 19 | 17.03–24.03 | 529 | 2340 | 23 | 1463 |
| **Sólheimaj.** | 02.07–06.07 | 521 | 3277 | 16 | 17.02–23.02 | 921 | 2819 | 33 | 1442 |
| **Þingvellir** | 16.06–22.06 | 3608 | 17,612 | 20 | 09.02–15.02 | 1921 | 7990 | 24 | 5529 |
| **Total** | | 12,199 | 48,391 | 25 | | 6758 | 18,992 | 36 | 18,957 |

* Number of visitors during data collection (Þórhallsdóttir and Ólafsson, unpublished data retrieved from authors).

A population-based probability sampling strategy such as that used in this study should allow for generalizability to the target population. Although there is potential for underrepresentation of certain groups of tourists due to various non-response biases, the data from the survey were not weighted. The reason for this is twofold; firstly the demographic composition of these large samples are in line with official data about nationalities of tourists arriving in Iceland [57] and secondly, limited official data exist about other demographic variables, such as the gender and age groups of tourists visiting particular areas. In fact, one of the purposes of this large dataset was to uncover more knowledge about these variables.

The questionnaires were available in English, German, French, and Icelandic. They contained 30 questions (some with sub-questions) but not all are presented in this paper. Respondents took 15–20 min to complete the questionnaires. The questions can be grouped into three categories:

1.  Visitor characteristics. Including age, gender, place of residence, party/group size.
2.  Activities/behavior. Activities while in the area, length of stay, travel time, time spent on site, time spent hiking, travel mode, accommodation type, past visits, and recreational use patterns.
3.  Attitude/motivation. The area's attractiveness, attitudes towards naturalness of the area, appropriate facilities, opinion and expectations on number of visitors and vehicles, satisfaction (with the biophysical condition, service, and infrastructure) as well as the influencing factors in their decision to visit the particular sites.

Most of the attitude/motivation questions were distributed on a five-point Likert scale ranging from, for example, 1 = very unsatisfied and 5 = very satisfied. The means for both winter and summer, were calculated for each location. Due to the large sample sizes and randomization in its collection, normality of the sampling distribution was assumed [69] and means were compared using independent t-tests to discover whether there was a statistically significant difference depending on the season. In the following analyses, a significance level of 0.05 is used (i.e., if $p < 0.05$ it is concluded that statistically significant differences exist between summer and winter visitors).

## 4. Results

### 4.1. Demographics and Travel Behavior

The majority of the tourists in the sample were international visitors (94%) and most (88%) of them were visiting Iceland for the first time. A high degree of seasonality can be observed in the demographic composition of travelers: in winter, almost half (43%) of the tourists were from Britain, while British visitors only accounted for 9% during the summer. Germans were the largest group in the summer (22%), but there were relatively few (8%) in the winter. Visitors from North America comprised about 16% of all visitors both in winter and summer. This division is similar to the official data for the composition of international visitors to Iceland [57]. The average age of travelers was

higher (44.7 years) and more evenly distributed during the summertime, while in winter the mean age was 38.0 years, with 54% of all guests 35 years or younger.

Mode of travel and accommodation also differed between seasons: in winter, the majority (64%) of the sample traveled by bus, while 27% used rental cars. Conversely, in the summer more (43%) used rental cars than bus travel (37%). Also, about 26% of tourists were traveling in an organized group in the summer, while a mere 18% were doing so in the winter. Most winter guests stayed in hotels (77%), and to a lesser degree hostels (15%), but in the summer the proportion staying in hotels was down to 59% with many opting to stay at campsites (18%) and farm-house accommodations (10%).

### 4.2. Visitors' Perception of the Place

All the destinations are perceived very positively with regard to their environmental qualities (i.e., for being very beautiful, natural, and clean) but somewhat less for being quiet. The areas were also perceived as safe and accessible. There is however considerable seasonality in tourists' perception of the areas. For instance, all the destinations, except Seltún and Sólheimajökull, were perceived to be more beautiful in the winter than in the summer and all except Sólheimajökull were considered quieter in the winter. Also, the three most visited destinations, Geysir, Jökulsárlón, and Þingvellir, were perceived as less natural in the summer than in the winter.

Jökulsárlón and Hraunfosar were considered less clean in summer than in winter, while Geysir and Seltún were considered cleaner in the summer. Generally, the research areas were considered more accessible and safer in the summer, but while Geysir, Þingvellir, and Djúpalónssandur were perceived as safer in the summer than in the winter, Hraunfossar was considered safer in the winter (Table 2).

**Table 2.** Visitors' perception of the environment in summer and winter.

| | Ugly–Beautiful | | | | Man-Made–Natural | | | | Dirty–Clean | | | |
|---|---|---|---|---|---|---|---|---|---|---|---|---|
| | Summer | Winter | *t* | *p* | Summer | Winter | *t* | *p* | Summer | Winter | *t* | *p* |
| **Djúpalónss.** | 4.78 | 4.88 | −2.864 | 0.005 | 4.66 | 4.66 | 0.073 | 0.942 | 4.66 | 4.62 | 0.613 | 0.540 |
| **Geysir** | 4.53 | 4.59 | −2.787 | 0.005 | 4.08 | 4.38 | −11.257 | <0.001 | 4.57 | 4.50 | 3.105 | 0.002 |
| **Hraunfossar** | 4.76 | 4.86 | −3.182 | 0.002 | 4.44 | 4.50 | −1.140 | 0.254 | 4.66 | 4.74 | −2.185 | 0.029 |
| **Jökulsárlón** | 4.77 | 4.86 | −3.224 | 0.001 | 4.46 | 4.66 | −5.160 | <0.001 | 4.44 | 4.67 | −6.501 | <0.001 |
| **Seltún** | 4.66 | 4.68 | −0.454 | 0.650 | 4.42 | 4.45 | −0.621 | 0.534 | 4.67 | 4.52 | 3.619 | <0.001 |
| **Sólheimaj.** | 4.79 | 4.84 | −1.524 | 0.128 | 4.63 | 4.70 | −1.878 | 0.061 | 4.59 | 4.63 | −0.952 | 0.341 |
| **Þingvellir** | 4.74 | 4.79 | −2.424 | 0.015 | 4.36 | 4.56 | −8.242 | <0001 | 4.68 | 4.66 | 1.091 | 0.275 |
| **All locations** | 4.70 | 4.72 | −2.866 | 0.004 | 4.36 | 4.51 | −11.855 | <0.001 | 4.60 | 4.59 | 1.082 | 0.279 |
| | Unsafe–Safe | | | | Inaccessible–Accessible | | | | Loud–Quiet | | | |
| | Summer | Winter | *t* | *p* | Summer | Winter | *t* | *p* | Summer | Winter | *t* | *p* |
| **Djúpalónss.** | 4.42 | 4.19 | 2.857 | 0.004 | 4.49 | 4.27 | 2.698 | 0.008 | 4.52 | 4.66 | −2.237 | 0.026 |
| **Geysir** | 4.49 | 4.27 | 8.955 | <0.001 | 4.49 | 4.20 | 12.015 | <0.001 | 3.52 | 4.10 | −18.914 | <0.001 |
| **Hraunfossar** | 4.35 | 4.47 | −2.609 | 0.009 | 4.50 | 4.53 | −0.817 | 0.414 | 4.19 | 4.74 | −13.271 | <0.001 |
| **Jökulsárlón** | 4.54 | 4.48 | 1.464 | 0.143 | 4.38 | 4.45 | −1.601 | 0.110 | 3.64 | 4.45 | −17.158 | <0.001 |
| **Seltún** | 4.37 | 4.38 | −0.045 | 0.964 | 4.53 | 4.43 | 2.333 | 0.020 | 4.41 | 4.58 | −3.981 | <0.001 |
| **Sólheimaj.** | 4.23 | 4.24 | −0.149 | 0.882 | 4.08 | 4.16 | −1.522 | 0.128 | 4.47 | 4.48 | −0.193 | 0.847 |
| **Þingvellir** | 4.58 | 4.37 | 8.329 | <0.001 | 4.52 | 4.32 | 8.593 | <0.001 | 4.20 | 4.40 | −7.561 | <0.001 |
| **All locations** | 4.48 | 4.33 | 11.624 | <0.001 | 4.46 | 4.28 | 13.769 | <0.001 | 3.99 | 4.35 | −23.496 | <0.001 |

Means calculated from a five-point Likert scale. For example, 1 = very ugly and 5 = very beautiful.

### 4.3. Visitor Satisfaction

Visitors' satisfaction regarding their stay in the area, nature, service, and the different types of infrastructure (restrooms, paths, parking areas, marking of places of interest, and signs/signboards) was generally high (Table 3). In all locations, the natural environment is what visitors were most satisfied with, both in summer and winter. Overall, summer and winter visitors were equally satisfied with the tour in general, except at Hraunfossar and Sólheimajökull, where satisfaction with the tour in general was significantly lower in the summer. Satisfaction with the natural environment was generally higher in winter than in the summer, especially at Hraunfossar, Jökulsárlón, and Sólheimajökull. In the summer, visitors were generally more satisfied with paths and less satisfied with the service than in winter—although for a couple of places the reverse is true. For instance, at the busy destinations of Geysir, Þingvellir, and Jökulsárlón, visitors were more satisfied with the service in the winter,

while at Djúpalónssandur and Seltún they were more satisfied in the summer. At Geysir and Þingvellir, satisfaction with infrastructure was generally higher during summer, while at Jökulsárlón it was higher in winter.

**Table 3.** Visitors' satisfaction with nature, tour, service, and different infrastructure.

| | Tour in General | | | | Nature | | | | Paths | | | |
|---|---|---|---|---|---|---|---|---|---|---|---|---|
| | Summer | Winter | *t* | *p* | Summer | Winter | *t* | *p* | Summer | Winter | *t* | *p* |
| Djúpalónss. | 4.29 | 4.33 | −0.562 | 0.574 | 4.68 | 4.73 | −0.668 | 0.504 | 4.28 | 4.02 | 3.270 | 0.001 |
| Geysir | 4.17 | 4.19 | −0.866 | 0.387 | 4.56 | 4.57 | −0.522 | 0.602 | 4.05 | 3.99 | 2.178 | 0.029 |
| Hraunfossar | 4.23 | 4.45 | −4.106 | <0.001 | 4.60 | 4.80 | −4.807 | <0.001 | 4.18 | 4.24 | −1.086 | 0.278 |
| Jökulsárlón | 4.25 | 4.25 | 0.010 | 0.992 | 4.70 | 4.80 | −3.135 | 0.002 | 3.90 | 4.01 | −2.362 | 0.018 |
| Seltún | 4.27 | 4.32 | −1.065 | 0.287 | 4.60 | 4.67 | −1.927 | 0.054 | 4.23 | 4.15 | 1.510 | 0.131 |
| Sólheimaj. | 4.23 | 4.36 | −3.005 | 0.003 | 4.69 | 4.78 | −2.206 | 0.028 | 4.17 | 4.27 | −2.025 | 0.043 |
| Þingvellir | 4.27 | 4.25 | 0.967 | 0.333 | 4.67 | 4.68 | −0.562 | 0.574 | 4.35 | 4.27 | 3.316 | 0.001 |
| All locations | 4.24 | 4.26 | −1.785 | 0.071 | 4.64 | 4.67 | −2.998 | 0.003 | 4.17 | 4.14 | 2.090 | 0.037 |

| | Parking | | | | Service | | | | Marking of Places of Interest | | | |
|---|---|---|---|---|---|---|---|---|---|---|---|---|
| | Summer | Winter | *t* | *p* | Summer | Winter | *t* | *p* | Summer | Winter | *t* | *p* |
| Djúpalónss. | 4.09 | 4.16 | −0.722 | 0.471 | 3.85 | 3.65 | 2.011 | 0.046 | 4.10 | 4.08 | 0.282 | 0.778 |
| Geysir | 4.15 | 3.96 | 8.087 | <0.001 | 4.02 | 4.17 | −6.624 | <0.001 | 3.97 | 3.88 | 3.220 | 0.001 |
| Hraunfossar | 3.96 | 4.33 | −6.447 | <0.001 | 3.66 | 3.67 | −0.188 | 0.851 | 3.92 | 4.02 | −1.748 | 0.081 |
| Jökulsárlón | 3.81 | 4.15 | −7.789 | <0.001 | 3.83 | 3.99 | −3.757 | <0.001 | 3.70 | 3.97 | −5.887 | <0.001 |
| Seltún | 4.25 | 4.15 | 2.411 | 0.016 | 3.68 | 3.53 | 2.795 | 0.005 | 4.14 | 4.06 | 1.606 | 0.108 |
| Sólheimaj. | 3.98 | 3.97 | 0.114 | 0.910 | 3.94 | 3.89 | 1.026 | 0.305 | 3.74 | 3.78 | −0.829 | 0.407 |
| Þingvellir | 4.08 | 3.99 | 3.367 | 0.001 | 3.85 | 3.96 | −4.139 | <0,001 | 3.96 | 3.90 | 2.222 | 0.026 |
| All locations | 4.05 | 4.02 | 1.856 | 0.063 | 3.86 | 3.98 | −8.500 | <0.001 | 3.93 | 3.91 | 1.657 | 0.097 |

| | Signs | | | | Restrooms | | | | | | | |
|---|---|---|---|---|---|---|---|---|---|---|---|---|
| | Summer | Winter | *t* | *p* | Summer | Winter | *t* | *p* | | | | |
| Djúpalónss. | 4.04 | 4.04 | −0.059 | 0.953 | 3.82 | 3.10 | 6.246 | <0.001 | | | | |
| Geysir | 3.90 | 3.85 | 1.941 | 0.052 | 4.07 | 4.18 | −4.354 | <0.001 | | | | |
| Hraunfossar | 3.89 | 3.98 | −1.489 | 0.137 | 3.44 | 3.69 | −3.996 | <0.001 | | | | |
| Jökulsárlón | 3.63 | 3.81 | −3.791 | <0.001 | 3.56 | 3.92 | −7.520 | <0.001 | | | | |
| Seltún | 4.03 | 3.98 | 0.991 | 0.322 | 3.75 | 3.12 | 9.936 | <0.001 | | | | |
| Sólheimaj. | 3.66 | 3.74 | −1.408 | 0.159 | 3.36 | 3.14 | 3.188 | 0.001 | | | | |
| Þingvellir | 3.86 | 3.83 | 1.117 | 0.264 | 3.55 | 3.55 | −0.117 | 0.907 | | | | |
| All locations | 3.85 | 3.85 | 0.413 | 0.680 | 3.69 | 3.72 | −1.729 | 0.084 | | | | |

Means calculated from a five-point Likert scale where 1 = very unsatisfied and 5 = very satisfied.

*4.4. Attitudes towards Number of Visitors and Vehicles*

Generally, the number of visitors and vehicles at the research sites was perceived to be acceptable, although visitors were more likely to experience crowds during the summer than in the winter. In all areas except Sólheimajökull, the number of visitors in general was perceived to be higher in the summer than in the winter (Table 4). The greatest seasonality in perceived tourist numbers was experienced at Geysir, where respondents in the summer experienced higher numbers of visitors in general, tour groups, buses, foreign visitors, and cars than their winter traveling counterparts. Large seasonal differences in perceived tourist numbers were also felt in Jökulsárlón, Hraunfossar, and Þingvellir. However, tour groups seem to be prominent all year round at Geysir, Jökulsárlón, Sólheimajökull, and Þingvellir where tour groups are generally perceived to be "rather many" both during summer and winter.

**Table 4.** Attitudes towards number of visitors and vehicles in summer and winter.

| | Visitors in General | | | | Tour Groups | | | | Buses | | | |
|---|---|---|---|---|---|---|---|---|---|---|---|---|
| | Summer | Winter | t | p | Summer | Winter | t | p | Summer | Winter | t | p |
| Djúpalónss. | 3.07 | 2.94 | 2.393 | 0.017 | 3.14 | 3.03 | 1.953 | 0.052 | 3.13 | 2.94 | 2.787 | 0.006 |
| Geysir | 3.49 | 3.17 | 16.719 | <0.001 | 3.55 | 3.29 | 11.920 | <0.001 | 3.48 | 3.27 | 10.056 | <0.001 |
| Hraunfossar | 3.13 | 2.79 | 8.790 | <0.001 | 3.12 | 2.81 | 6.080 | <0.001 | 3.09 | 2.74 | 7.017 | <0.001 |
| Jökulsárlón | 3.45 | 3.13 | 8.884 | <0.001 | 3.51 | 3.33 | 4.310 | <0.001 | 3.48 | 3.17 | 7.725 | <0.001 |
| Seltún | 3.05 | 2.93 | 4.009 | <0.001 | 3.04 | 2.98 | 1.471 | 0.142 | 2.98 | 3.00 | −0.490 | 0.624 |
| Sólheimaj. | 3.06 | 3.07 | −0.220 | 0.826 | 3.24 | 3.30 | −1.633 | 0.103 | 3.20 | 3.19 | 0.310 | 0.757 |
| Þingvellir | 3.24 | 3.10 | 8.262 | <0.001 | 3.31 | 3.26 | 2.705 | 0.007 | 3.25 | 3.19 | 3.255 | 0.001 |
| All locations | 3.29 | 3.09 | 20.593 | <0.001 | 3.34 | 3.23 | 9.664 | <0.001 | 3.30 | 3.17 | 10.831 | 0.001 |
| | Foreign Visitors | | | | Cars | | | | Icelandic Visitors | | | |
| | Summer | Winter | t | p | Summer | Winter | t | p | Summer | Winter | t | p |
| Djúpalónss. | 3.07 | 2.95 | 2.139 | 0.033 | 3.01 | 2.89 | 2.331 | 0.020 | 2.64 | 2.60 | 0.435 | 0.664 |
| Geysir | 3.43 | 3.14 | 15.521 | <0.001 | 3.30 | 2.99 | 18.199 | <0.001 | 2.75 | 2.70 | 1.922 | 0.055 |
| Hraunfossar | 3.12 | 2.83 | 7.008 | <0.001 | 3.08 | 2.82 | 7.588 | <0.001 | 2.71 | 2.49 | 4.279 | <0.001 |
| Jökulsárlón | 3.37 | 3.15 | 6.362 | <0.001 | 3.38 | 3.01 | 11.396 | <0.001 | 2.80 | 2.48 | 7.233 | <0.001 |
| Seltún | 3.08 | 2.95 | 3.802 | <0.001 | 3.03 | 2.91 | 4.652 | <0.001 | 2.53 | 2.59 | −1.244 | 0.214 |
| Sólheimaj. | 3.06 | 3.11 | −1.747 | 0.081 | 3.02 | 2.91 | 3.524 | <0.001 | 2.65 | 2.60 | 1.201 | 0.230 |
| Þingvellir | 3.20 | 3.12 | 4.446 | <0.001 | 3.10 | 2.91 | 11.893 | <0.001 | 2.79 | 2.62 | 7.350 | <0.001 |
| All locations | 3.25 | 3.10 | 15.793 | <0.001 | 3.18 | 2.94 | 26.736 | <0.001 | 2.74 | 2.63 | 8.613 | <0.001 |

Means calculated from a five-point Likert scale where 1 = too few visitors/vehicles and 5 = too many visitors/vehicles.

## 4.5. Effects of Tourism on the Environment

Visitors did not observe many negative effects of tourism on the environment, although it was more likely to occur in the summer rather than the winter. The most observed issue was with respect to footpath erosion, and the place it was most noticed was Geysir. Visitors at Geysir, Jökulsárlón, and Þingvellir, the most visited destinations, as well as at Djúpalónssandur, noticed more erosion in the summer than in winter. Damaged vegetation was seen more in summer than in winter at all destinations and the same went for damaged geological formations, except at Sólheimajökull where there was no statistical difference between seasonal perceptions. Litter was hardly noticed, although there was significantly more in summer than in winter, especially at the three most visited destinations of Jökulsárlón, Geysir, and Þingvellir (Table 5).

**Table 5.** Visitors' notion of the effects of tourism on the environment in summer and winter.

| | Erosion of Paths | | | | Damaged Vegetation | | | |
|---|---|---|---|---|---|---|---|---|
| | Summer | Winter | t | p | Summer | Winter | t | p |
| Djúpalónssandur | 2.05 | 1.67 | 3.690 | <0.001 | 1.73 | 1.44 | 3.420 | 0.001 |
| Geysir | 2.22 | 1.80 | 13.258 | <0.001 | 1.93 | 1.47 | 16.613 | <0.001 |
| Hraunfossar | 1.97 | 2.01 | −0.641 | 0.522 | 1.71 | 1.55 | 2.809 | 0.005 |
| Jökulsárlón | 2.08 | 1.80 | 4.819 | <0.001 | 1.79 | 1.50 | 6.464 | <0.001 |
| Seltún | 2.10 | 2.15 | −0.696 | 0.487 | 1.76 | 1.59 | 3.144 | 0.002 |
| Sólheimajökull | 2.08 | 2.11 | −0.407 | 0.684 | 1.62 | 1.47 | 2.881 | 0.004 |
| Þingvellir | 1.81 | 1.57 | 8.455 | <0.001 | 1.70 | 1.29 | 17.277 | <0.001 |
| All locations | 2.02 | 1.82 | 12.273 | <0.001 | 1.77 | 1.43 | 24.323 | <0.001 |
| | Damaged Geological Formations | | | | Litter | | | |
| | Summer | Winter | t | p | Summer | Winter | t | p |
| Djúpalónssandur | 1.47 | 1.34 | 1.965 | 0.051 | 1.51 | 1.42 | 1.210 | 0.227 |
| Geysir | 1.74 | 1.47 | 10.797 | <0.001 | 1.63 | 1.53 | 3.727 | <0.001 |
| Hraunfossar | 1.57 | 1.43 | 3.083 | 0.002 | 1.41 | 1.36 | 0.945 | 0.345 |
| Jökulsárlón | 1.70 | 1.50 | 4.463 | <0.001 | 1.73 | 1.58 | 3.064 | 0.002 |
| Seltún | 1.58 | 1.47 | 2.363 | 0.018 | 1.53 | 1.55 | −0.391 | 0.696 |
| Sólheimajökull | 1.62 | 1.56 | 1.138 | 0.255 | 1.52 | 1.42 | 2.412 | 0.016 |
| Þingvellir | 1.44 | 1.28 | 7.835 | <0.001 | 1.45 | 1.35 | 4.884 | <0.001 |
| All locations | 1.59 | 1.43 | 12.938 | <0.001 | 1.55 | 1.46 | 7.179 | <0.001 |

Means calculated from a five-point Likert scale where 1 = not at all and 5 = very much.

## 5. Discussion

### 5.1. Tourism Seasonality in Iceland

Previous studies on seasonality in natural areas have tended to focus on differences in tourist numbers rather than on how visitors' perceptions may be related to seasonal landscapes. Conversely, this study examined whether there are differences in perception and satisfaction among tourists between summer and winter at seven popular nature destinations in Iceland.

Despite dramatic improvements in seasonality in the capital region, Iceland as a whole continues to be, like most polar destinations, a single peak destination, characterized by summer visitation to the various nature destinations [70]. The reasons for the capital region's reduced seasonality include state-sponsored marketing endeavors and foreign airlines' inclusion of Iceland as a destination. However, because of the shorter average stays during winter and the remoteness and accessibility of the northern and eastern parts of the country, the increased off-season tourism has had limited effects on areas further away from the capital. As a result, tourism operators in rural Iceland may struggle with limited return on investment [30,31], and difficulties with hiring and retaining staff [32,33]. During the off-season service is also limited and many businesses do not operate throughout the year. Nevertheless, a certain level of seasonality has to be expected since, as Page and Hall [20] pointed out, peripheral destinations relying on natural and climatic elements will tend to have greater levels of seasonality than cities.

Visits to a destination like Iceland, and other polar regions, have until recently been perceived as climatically marginal [70,71]. That is mostly due to what Butler [1] refers to as "natural causes", such as unfavorable weather conditions and limited daylight hours. Mansfeld, Freundlich, and Kutiel [72] have also shown that natural elements (e.g., weather and daylight), have a great influence on tourism. Interestingly however, the negative weather conditions' effects on off-season travel to Iceland seem to be diminishing in recent years as the percentage of tourists naming nature as a main reason for traveling to the country during winter has been gradually increasing. Nature has historically been the main reason international guests visit Iceland, with over 80% of summer visitors consistently naming nature as the main reason for visiting [62]. The percentage of winter tourists coming to Iceland because of its nature has increased from 48% in the winter of 1998/1999 to 74% in 2015/2016 [60]. Thus, the winter climate seems to play a lesser role in tourists' ability to enjoy Icelandic nature in recent years. The abundance of travel information as Iceland has become more popular may have helped to temper some myths about the harshness of winter conditions, while the winter landscape has increasingly come to be seen as a positive aspect of the tourist experience.

The summer when the study was conducted was wetter than average, which might have reduced satisfaction in the summer. The weather in the winter the year the study was conducted was also very variable. As a result, road conditions can affect visitors' geographical travel patterns. The heavily visited destinations, like Þingvellir and Geysir, are close to the capital area, between one and two hours driving time and with a relatively good road surface. As Butler [1] and Hall and Johnston [28] have pointed out, the limited accessibility of remote areas during off-season periods can be a major hindrance for a change in visitation patterns. That is most likely the case for the more rural areas of Iceland that receive limited visits in the winter. Hence, seasonality not only has a sequential dimension but also a spatial one [1,25].

This research shows several key differences in the perception and satisfaction of summer and winter tourists at these nature destinations, which may reveal some key reasons for traveling during the off-season. These can serve as a basis for further marketing of winter tourism outside of Iceland's capital.

### 5.2. Seasons, Satisfaction, and Perception

This study indicates that some aspects of satisfaction and perception are quite different between the two seasons and that the experience of visitors in winter is generally more positive than in the summer. Although there was little difference in the overall satisfaction between summer and winter,

satisfaction with the natural environment, hiking paths and, to a larger extent service was generally higher in the winter. At two places, Djúpalónssandur and Seltún, satisfaction with services was lower during the winter—reflecting the fact that none was provided in the area in the winter. This is in line with what Bar-On [16] refers to as the social and personal costs of seasonality, such as a lower quality of service in the high season. Furthermore, environmental qualities (e.g., beauty, naturalness, cleanliness, and quietness) are generally experienced less positively in the summer than the winter. Summer visitors also observe more of all kinds of negative environmental impacts of tourism, such as littering, erosion of paths, and damage to vegetation and geological formations, than in the winter. On the other hand, some of the destinations are considered less safe as well as being less accessible in the winter.

### 5.3. Crowding and Seasonality

It has long been recognized that the concentration of visitors at tourist destinations during the peak period can reduce visitor satisfaction due to overcrowding [1]. However, this issue is gaining increased attention given concerns over "overtourism" and sustainable tourism [73]. In Iceland, visitors' perception of overcrowding in the summer at the most popular destinations was quite high, which stresses the importance of better distribution throughout the year in order to aim for reducing the pressure of tourism on the environment, as well as providing positive tourist experiences. In the summer, 36% of visitors at Geysir thought that the number of tourists was above what they would consider suitable, while 35% at Jökulsárlón and 20% at Þingvellir were of the same opinion. Group travelers were more often considered too many than "the general tourist", as about 39% of visitors at Geysir, 38% of visitors at Jökulsárlón, 26% at Þingvellir, and 24% at Sólheimajökull considered their number being above suitable.

In recent years the single peak visitation pattern has been changing as there has been a substantial increase in off-season travel to the country. Tourists visit the same nature destinations in the southwest of the country in winter as in summer. Even though visitors' perception of crowding in the summer was more prevalent than in winter at all the destinations, crowding was also experienced at the most popular destinations in the winter. A total of 16% of visitors at Jökulsárlón, 15% at Geysir, and 13% at Sólheimajökull and Þingvellir felt the number of tourists at the sites was unsuitable in winter. Those who experienced crowding in winter were thus quite many, compared to the difference in actual visitor numbers in winter and summer. As shown by Þórhallsdóttir and Ólafsson [65], Þingvellir had over 100,000 visitors in July but "only" 31,000 in February, while Jökulsárlón had 88,000 in July and 14,000 in February. The proportionally high perception of too many tourists in winter can partly be explained by visitors having different expectations for the two seasons. Furthermore, the benefits sought for each season can differ, resulting in destinations attracting different markets in winter and summer [21,74]. This may be part of the reason for the different demographics and travel behaviors between the seasons revealed by this research.

### 5.4. Limits of Current Research

This study only looked at visitors' perception of the situation at locations for specific perceptual factors. Even though the number of visitors during the month the survey was conducted is known, the exact number of visitors visible to the respondents while they experienced the area is not known.

The destinations are different in ways other than in numbers (e.g., the different landscapes at each location mean that they have different visual fields which may influence relative visibility of other people at the sites). Service and other facilities are also different which can affect visitor satisfaction.

The weather is an important factor that affects visitors' satisfaction. However, due to the frequently changing weather in Iceland the situation during the time each individual stayed in the area and answered the questionnaire is not recorded. Some nationalities could potentially be underrepresented in the survey. Although it covered most requirements to have the questionnaires in English, German,

French, and Icelandic, it would have been beneficial to have versions in more languages like Mandarin, Spanish, and Italian.

A few groups did not respond at all because the tour guides could not (or were unwilling to) give their clients the time needed. This problem was mostly avoided as the travel guide, or bus driver, of the groups who did not have time to answer the questionnaire on site was asked to take the questionnaire onto the bus and return them after completion via mail. Nevertheless, group tours may possibly be underestimated in the sample. The proportion of the sample passing through during periods of peak daily visitation is probably underrepresented as the number of interviewers distributing the questionnaire on site was the same during all times of the day. That means that during "rush hour" proportionally fewer replied to the questionnaire than during other times of the day. Therefore, if the total population had been interviewed the problem of overcrowding might have been even larger than recorded in this study.

## 6. Implications and Conclusions

It is vital to know how visitors perceive the characteristics of a destination in order to determine its competitive position [1,3,75]. The first implication of the study is thatthe problem of overcrowding at the most popular tourist destinations in Iceland is already a great concern for visitors, both in the summer and winter. Despite great success in extending the tourist season in Iceland, the tolerance level for overcrowding seems to be lower in the winter than in the summer, since even though there are far fewer tourists in the winter, many respondents still consider the number of tourists too high. This may be due to winter visitors expecting even fewer tourists. Strategies to reduce the negative impacts of overcrowding can include an increase in tariffs on tourism, the geographical redirection of the demand to other destinations, the creation of new attractions, a development of infrastructure at new destinations, or increased access to more remote areas through, for example, better roads or international airports in other parts of the country that are capable of receiving more traffic and visitors than they do now.

The second implication of the study is that its perceptual analysis identifies some of the places that create weaknesses and threats for tourism in Iceland as a nature-based tourism destination. Some of these may be inevitable due to the uneven geographical distribution of natural phenomena in relation to centers of tourist accommodation. Others, however, give a clear message to managers and planners regarding these destinations and their need to be reorganized as the flow of traffic, as well as the development of infrastructure and services, are better managed. The results should also be used by managers to see what the main problems are in each location, depending on the season, and what needs to be changed or modified to increase visitors' satisfaction and thus improve the competitive advantage.

A third implication may be the need to apply strategies to reduce seasonality in areas of the country where it is most prominent. Although this could eventually become a double-edged sword, winter can be marketed as a time when travelers may get more out of their stay in terms of natural beauty, quietness, and lack of crowds. In spite of limited daylight hours and scarce vegetation, as well as colder temperatures, tourists are more satisfied with the natural environment in the winter than in the summer. Some visitors may come from parts of the world where such unique conditions are never present, making a winter stay in Iceland a more "adventurous" experience, while such experiences may even become more attractive given the consequences of climate change [52]. This premise can be used in marketing material in hopes of encouraging people to stay longer in rural areas, closer to quieter destinations. Nevertheless, identifying the "ideal" degree of seasonality for a destination remains a relevant and important question, depending as it does not only upon a range of different stakeholder concerns in the destination, but also the motivations and perceptions of the visitor, businesses, and host communities, as well as the resilience and robustness of locations with respect to the environmental pressures of tourism [76]. While there is never a perfect answer to such questions, research such as that conducted for the present study, points to the need to better understand visitor markets by season

in order to not only to be able to meet the needs of the market, but also to better sustain the natural resources upon which the tourist experience is based [75].

**Author Contributions:** Writing, review, and editing—A.D.S., C.M.H. and Þ.S.

**Funding:** This research was funded by the Icelandic Tourist Board.

**Acknowledgments:** We thank the Icelandic Tourist Board and its director at the time Ólöf Ýr Atladóttir for the initiative in financing this research. We also thank Rögnvaldur Ólafsson and Gyða Þórhallsdóttir for providing information on the number of visitors at the research areas, the RHA-University of Akureyri Research Centre for scanning the questionnaires into a digital format, and Edita Tverijonaite for making the map. Finally, we thank the research assistants who distributed the questionnaires at the destinations for their assistance: Gísli Bjarki Guðmundsson, Gyða Þórhallsdóttir, Ingunn Árnadóttir, Kristján Smárason, Margrét Sævarsdóttir, Silja Gunnarsdóttir, and Zsófia Cságoly.

**Conflicts of Interest:** The authors declare no conflict of interest.

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
