# Peer review of "Senses by Seasons: Tourists’ Perceptions Depending on Seasonality in Popular Nature Destinations in Iceland"

_sustainability, doi:10.3390/su11113059_

Round 1

Reviewer 1 Report

Thank you for the opportunity to review this paper. Overall, I found this study to be clearly written and original enough to warrant publication.

Some vital information is missing from the method including:

Please outline what sampling strategy was used. Line 223 states that as many visitors as possible were approached. This sounds like a convenience sampling strategy. Or was there some kind of randomization?

Were postage-paid envelopes distributed to all bus drivers or who was in charge of mailing/paying?

18,957 questionnaires were collected. Are you able to report on how many questionnaires in total were distributed?

Please outline how data were analysed in the method section, for example why was the t-test chosen to analyse the data? If no randomization took place during data collection then non-parametric analysis would make more sense, i.e. Mann-Whitney Test.

There are some minor issues in-text which are quickly fixed by addressing some of the points below.

Line 37-39: Sentence includes ‘demand’ three times.

Line 41: Marked by …. Incomplete sentence?

Line 62: You go on to explain what non-peak is. It might be worthwhile to add an example of a double peak destination for your readers.

Line 137: Airfares (not air-fares)

Line 184: The exception is …. So can all 7 destinations be reached via daytrip or not? The sentence is rather awkward and may require some rewording.

Line 208: Space after Jokulsarlon

Line 225: questionnaires (plural)

Line 255: ranked might be a better word than ‘distributed’

Line 290: but although – use either ‘but’ or ‘although’ (both sound awkward)

Line 293: Please check ALL your table headings. I think Table 7 should be Table 2. Maybe Table 2 should be Table 4 (???).

Line 490: … the survey was (not is) conducted is known,…

Line 485: … for scanning them… What does ‘them’ stand for? The surveys?

Thank you! I hope you find these comments helpful. 

Author Response

Response to Reviewer 1 Comments

We thank the reviewer for very good and constructive comments which we feel have really improved the quality of the paper.

·         We have had language and style re-checked.

·         Please outline what sampling strategy was used. Line 223 states that as many visitors as possible were approached. This sounds like a convenience sampling strategy. Or was there some kind of randomization?

We have attempted to improve the method chapter.

o   We have outlined better what sampling strategy was used by adding the following text:

The target population of this study were all visitors at popular nature destinations in Iceland. Devising an effective sampling frame can be problematic in on-site tourism research in natural areas. Accessing the entire population is extremely difficult as the study area is often very large or parts of it are inaccessible. It was obviously unrealistic to gather information from the entire visitor population in large natural areas, and therefore a sample had to be taken. Seven popular nature destinations were selected that represent a wide range of environments and use levels. Data was collected by one to five interviewers, depending on the destination’s previously reported popularity [see 62]. The interviewers stayed at each destination for a week. A week-long stay was chosen at each location as some visits follow a day-of-the-week pattern. For example, the number of Icelandic visitors is higher at the weekend, while some organized group tours have fixed days of arrival each week. A self-completion survey was given out during the day approximately between 9 am and 7 pm. The utilisation of self-completion questionnaires required less staff to gather the data and made it possible to get answers from tourists on buses. Primarily, a simple random sampling was used where the interviewers attempted to approach as many visitors as possible. However, organized tourist groups often do not have time to answer the questionnaire on site as their visit is part of a set tour timetable. They can therefore easily be under represented by a simple random sample approach. Therefore, in order to avoid bias, a stratified sampling was used for tour groups with travel guides or bus drivers being asked to distribute the questionnaire among the visitors on the bus. Those who agreed were given postage-paid envelopes and were asked to return them via mail after completion.

·         Were postage-paid envelopes distributed to all bus drivers or who was in charge of mailing/paying?

o   We have added that postage-paid envelopes were distributed to bus drivers and tour guides.

·         18,957 questionnaires were collected. Are you able to report on how many questionnaires in total were distributed?

o   Unfortunately, we cannot report how many questionnaires in total were distributed. There were quite many staff members in the project and not all of them realized the importance of that information and we lost track of that counting.

·         Please outline how data were analysed in the method section, for example why was the t-test chosen to analyse the data? If no randomization took place during data collection then non-parametric analysis would make more sense, i.e. Mann-Whitney Test.

o   We have outlined better how data were analyzed by adding:

Because of the large sample sizesand randomization in its collection, normality of the sampling distribution was assumed [66] and means were compared using independent t-tests to discover whether there was a statistically significant difference depending on season.

Some minor issues:

·         Line 37-39: Sentence includes ‘demand’ three times.

The sentence is now:

“Seasonal demand occurs when the pattern of demand over time is characterised by regular fluctuations that are relatively consistent with the pattern of supply, although excess capacity to supply the product will exist at certain times of the year.”

·         Line 41: Marked by …. Incomplete sentence?

The sentence is now:

‘A state in which the current timing pattern of demand is marked by seasonal or volatile fluctuations that depart from the timing pattern of supply’

·         Line 62: You go on to explain what non-peak is. It might be worthwhile to add an example of a double peak destination for your readers.

We have added:

“Double peak destinations experience two periods of higher demand than other times of the year. This situation is often the case for alpine resorts that experience peak demand in summer and winter [18]”

·         Line 137: Airfares (not air-fares)

o   Thank you!

·         Line 184: The exception is …. So can all 7 destinations be reached via daytrip or not? The sentence is rather awkward and may require some rewording.

Thank you! We have changed the text to:

“All seven research areas are located in the south-west of the country and can be reached on a daytrip from the capital area. Jökulsárlón is located furthest away from Reykjavík (379 km) and is about a 12-15 hour round daytrip from Reykjavík.”

·         Line 208: Space after Jokulsarlon

o   Thank you!

·         Line 225: questionnaires (plural)

o   Thank you!

·         Line 25: ranked might be a better word than ‘distributed’

o   Here we really don’t agree with the reviewer and prefer to keep it as it is.

“The average age of travellers is higher (44.7 years) and more evenly distributed during the summertime, while in winter the mean age is 38.0 years, with 54% of all guests 35 years or younger”.

·         Line 290: but although – use either ‘but’ or ‘although’ (both sound awkward)

o   We completely agree and took “although” out. Thank you!

·         Line 293: Please check ALL your table headings. I think Table 7 should be Table 2. Maybe Table 2 should be Table 4 (???).

o   Thank you for pointing this out!

·         Line 490: … the survey was (not is) conducted is known,…

·         Thank you!

·         Line 485: … for scanning them… What does ‘them’ stand for? The surveys?
Thank you for pointing this out! We have changed the sentence to: “We also thank Rögnvaldur Ólafsson and Gyða Þórhallsdóttir for providing information on numbers of visitors at the research areas, the RHA - University of Akureyri Research Centre for scanning the questionnaires into a digital format and Edita Tverijonaite for making the map.”

Reviewer 2 Report

This is a really great, informative, and well-written paper. I recommend its acceptance after minor improvements. See my recommendations below.

1)    The relevance of this paper to sustainability issues should be well demonstrated in Abstract, Introduction, Discussion, and Conclusion.

2)    Section 6. Conclusion should exist!

3)    Lines 237, 292, 333: what is questioned in bold?

4)    Please, format citations and references according to the journal rules.

5)    Please, consider this work: https://www.sciencedirect.com/science/article/pii/S0261517713002185

6)    Please, think about the relevance of Icelandic tourism to climate change – something useful can be found here: https://journals.tdl.org/ertr/index.php/ertr/article/view/375

7)    As this work is about seasonality, two important papers published recently should be considered and cited: https://www.sciencedirect.com/science/article/abs/pii/S221197361930039X and https://www.sciencedirect.com/science/article/pii/S0261517718300657

Author Response

Response to Reviewer 2 Comments

We thank the reviewer for very good and constructive comments which we feel have really improved the quality of the paper.

·         We have had language and style re-checked.

·         We have attempted to support the conclusions better with the results

·         We have formatted citations and references according to the journal rules.

·         We have added the informative references the reviver pointed out to us.

·         We have empathized the relevance of this paper to sustainability issues in the manuscript. For example:

o   Abstract: Seasonality in visitor arrivals is one of the greatest challenges faced by tourist destinations. Seasonality is a major issue for sustainable tourism as it affects the optimal use of investment and infrastructure…

o   Introduction: Seasonality in tourism has substantial implications for the financial sustainability of businesses and the broader environmental, economic and social sustainability of destinations.

o   Discussion: However, this issue is gaining increased attention given concerns over ‘overtourism’ and sustainable tourism… AND While there is never a perfect answer to such questions, research such as that conducted for the present study, points to the need to better understand visitor markets by season in order not only to be able to meet the needs of the market but also the desire to better sustain the natural resources upon which the tourist experience is based.

We did      not add a new chapter with conclusions as according to Sustainability: ”Conclusions: This section is not      mandatory, but can be added to the manuscript if the discussion is      unusually long or complex.”

·         We did not understand the comment: Lines 237, 292, 333: what is questioned in bold?
